# Application of Natural Deep Eutectic Solvents in the Extraction of Quercetin from Vegetables

**DOI:** 10.3390/molecules24122300

**Published:** 2019-06-21

**Authors:** Yunliang Dai, Kyung Ho Row

**Affiliations:** Department of Chemistry and Chemical Engineering, Inha University, Incheon 402751, Korea; 22172312@inha.edu

**Keywords:** natural deep eutectic solvents (NADESs), fourier transform-infrared (FT-IR) spectroscopy, reverse phase high-performance liquid chromatography (RP-HPLC), quercetin

## Abstract

Quercetin is a phytochemical with disease prevention and health promotion activities that has attracted significant research attention. In this study, choline chloride and betaine-based natural deep eutectic solvents were prepared using a heating method. Their physical and chemical properties were also tested. Then, they were applied to extract quercetin from onion and broccoli with ultrasonic-assisted solid liquid method coupled with HPLC. Three factors (temperature, amount, and time) were considered for the optimization of the extraction assays. In the optimal conditions, the extraction recoveries were 88.91–98.99%, 88.45–99.01%, and 89.56–98.74% for quercetin, isorhamnetin, and kaempferol. Tailor-made natural deep eutectic solvents could be applied as sustainable and safe extraction media for biochemical applications.

## 1. Introduction

Quercetin (3, 5, 7, 3′, 4′-pentahydroxyflavone), a polar polyphenolic compound with a molecular formula of C_15_H_10_O_7_, is a natural flavonoid found abundantly in many edible vegetables [1,2]. Quercetin has attracted considerable attention because of its biologically important activities, such as anti-oxidant activity [3], anti-obesity effects [4], and cancer treatment potential [5,6]. The consumption of vegetables that are abundant in polyphenols has been suggested to decrease the risk of chronic diseases [7]. In some countries, quercetin is available as a dietary supplement with daily doses between 200 and 1200 mg [8]. Onions and wine have been reported to be rich sources of quercetin [9].

Deep eutectic solvents (DESs), generally tailor-made from two or three inexpensive and safe components that are capable of self-association, often through hydrogen bond interactions, to form a eutectic mixture with a melting point lower than that of each individual component. These solvents are new serial solvents or ionic liquid (IL) analogues that have been proposed to overcome some of the disadvantages of ionic solvents, including high melting point, large charge, and toxicity. Despite having many characteristics of ILs, DESs are a different category of solvent [10,11,12]. Since the first publication of DESs [13], a large number of new DESs have been obtained by diverse combinations of hydrogen bond donors (HBDs) and hydrogen bond acceptors (HBAs). Owing to their biodegradability, low cost, and lower toxicity [14], the most common DESs are based on choline chloride (ChCl)/betaine (Bet) as the HBA. ChCl and Bet have been combined with different HBDs, such as sugars [15], amides [16], alcohols [17], organic acids [18], and carbohydrates [19]. DESs have been applied widely in areas, such as solvents [20], bioengineering [21,22,23], nanotechnologies [24], separation of natural compounds [25,26,27], and new materials [28,29,30].

When DESs are prepared from natural origin components, they are defined further as natural deep eutectic solvents (NADESs) [31]. NADESs are mixtures of certain molar ratios of natural compounds, e.g., sugars, organic acids, amino acids, and organic bases, which are abundant in organisms. The aim of this study was to develop carbohydrate-based NADESs, examine their chemical properties and evaluate the extraction capacities of them. For this purpose, mixtures of different HBAs, i.e., ChCl and Bet, and nine natural carbohydrates as HBDs, i.e., meso-erythritol (Ery), d-(−)-ribose (Rib), xylitol (Xyl), d-(+)-glucono-1, 5-lactone (Glul), d-glucuronic acid (Glua), l-(−)-fucose (Fuc), d-mannitol (Mal), d-(+)-mannose (Man), and d-(+)-galactose (Gal) were used for the formation of NADESs. Because all the obtained NADESs are quite viscous, and considering that other DESs based on ChCl or Bet as the HBA consistently absorb a certain amount of water from the atmosphere, a 0.1 or 1 molar ratio of water was added to all ChCl or Bet based NADESs to lower their viscosities. The extraction abilities of the carbohydrate-based NADESs were measured and are illustrated in this study. Furthermore, the chemical structures were also characterized and discussed.

## 2. Results and Discussion

### 2.1. Characterization of the Prepared NADESs

#### 2.1.1. Fourier Transform-Infrared (FT-IR) Spectroscopy

Figure 1 presents FT-IR spectra of the prepared ChCl-based NADESs. The bands at 3260 cm^−1^ and 1210 cm^−1^ were assigned to hydroxyl groups (C-N^+^ symmetric stretching second ones). The peak at 2810 cm^−1^ was attributed to alkyl group, and the -CH_2_ being vibration at 1480 cm^−1^ was a prominent group detected in all ChCl-based DESs.

The major peaks for ChCl–Ery at 2920 cm^−1^ and 1414 cm^−1^ were assigned to O-H stretching and O-H bending vibrations, respectively (Figure 1a).

Figure 1b shows the spectrum of ChCl–Rib. The strong peak at approximately 3310 cm^−1^ was assigned to the stretching vibration of the O-H group, and the sharp peak at 1680 cm^−1^ was due to a five-membered ring. The combined effects of the C-N stretching vibration of ChCl and the C-O stretching vibration from the primary alcoholic hydroxyl group in Rib led to the formation of a band at approximately 1096 cm^−1^.

As shown in Figure 1c, the presence of ChCl–Xyl was confirmed by the vibrational bands at 2880 cm^−1^ and 1485 cm^−1^, which were assigned to alkyl groups.

Figure 1d presents the spectrum of ChCl–Fuc. The bands at 3500 cm^−1^ and 1210 cm^−1^ were assigned to the strong stretching and bending vibrations of C-H, respectively; and a peak at 1710 cm^−1^ was attributed to the C-O bending vibration that overlapped with a six-membered ring.

Figure 1e presents the spectrum of ChCl–Glul. The C-H stretching and bending vibrations were observed at 3260 and 1220 cm^−1^, and a sharp peak at 1750 cm^−1^ was assigned to a six-membered lactone group.

Figure 1f shows the spectrum of ChCl–Man, which was confirmed by the overlapped wide band from 3500 to 3000 cm^−1^ (O-H and N-H stretching vibrations), and a band at 1695 cm^−1^ due to a six-membered nitrogen heterocyclic ring.

Figure 1g presents the spectrum of ChCl–Gal. The major peaks were assigned to the following: a wide N-H and O-H stretching vibration at 3400–3020 cm^−1^, and characteristic peak for a six-membered ring at 1705 cm^−1^.

Figure 1h shows the spectrum of ChCl–Mal. The bands at 3310, 2885, and 1030 cm^−1^ were assigned to the O-H stretching vibration, C-H stretching vibration of a sp^3^-hybridized carbon, and O-H bending vibration of hydroxyl groups, respectively.

The spectrum for ChCl–Glua revealed bands at 1748 cm^−1^ and 1625 cm^−1^, which were assigned to a six-membered ring and C=O stretching vibration, respectively (Figure 1i).

Appendix A shows the FT-IR spectra of the prepared Bet-based NADESs. The peak at approximately 2200 cm^−1^ was assigned to the C=O-O group, and the other major information of vibrational bands was similar to that shown in Figure 1.

#### 2.1.2. Thermogravimetric Analysis (TGA)

TGA was conducted to examine the thermal stability of all the compounds applied to prepare NADESs, and the results were shown in Figure 2. For all components used to form the NADESs, the TGA curves showed a distinct two-stage degradation process with a major weight loss from 170 °C to 285 °C caused by an initial vaporization of carbohydrates, and a final weight loss between 285–400 °C at a slower rate due to complete decomposition.

As shown in Figure 2, Bet was more stable than ChCl, which was due to the difference in their melting points. Glua, Mal, and Xyl were the most stable of all the eutectic components analyzed, whereas Gal, Man, and Rib were the least stable. The most stable NADESs were those formed from HBDs with high thermal stabilities and boiling points. Therefore, HBDs with low volatility and high thermal stability should be considered when NADESs with high stabilities are required.

### 2.2. Selection Of The Extracting Method And Solvent

Compared to three other most commonly used extraction methods (stirring, heating, and stirring + heating) under the same conditions with methanol as the extraction solvent, ultrasonic-assisted solid liquid extraction was applied to the extraction process of quercetin with NADESs because of its high efficiency and simple operation (Figure 3). Further extraction conditions were optimized as described below.

Aqueous alcohols with different water contents are general solvents for plant extraction. In particular, methanol and its aqueous solutions are the most commonly used solvents. Although the differences were not significant, a comparison of 50%, 75%, and 100% methanol indicated 100% methanol to be the most effective. Therefore, 100% methanol was used as the extraction solvent to compare the extraction recoveries of various NADESs.

### 2.3. Selectivity of the NADESs

The extraction capacity of the ChCl-based NADESs (ChCl–Ery, ChCl–Rib, ChCl–Xyl, ChCl–Fuc, ChCl–Glul, ChCl–Man, ChCl–Gal, ChCl–Mal, and ChCl–Glua) and Bet-based NADESs (Bet–Ery, Bet–Rib, Bet–Xyl, Bet–Fuc, Bet–Glul, Bet–Man, Bet–Gal, Bet–Mal, and Bet–Glua) were studied. As shown in Figure 4, the extraction recoveries of the Bet-based NADESs were slightly higher than those of the ChCl-based NADESs. Bet–Man (ChCl–Man), Bet–Mal (ChCl–Mal), and Bet–Glua (ChCl–Glua) showed better recoveries than the other NADESs, which might be due to the differences in their spatial structures and chemical properties. Bet–Mal showed the best extraction recoveries under the optimal conditions: 95.75% for quercetin, 93.82% for isorhamnetin, and 91.71% for kaempferol.

### 2.4. Optimization of Quercetin Extraction

The extraction time is an important factor affecting solid liquid extraction, because it can have a direct influence on the distribution equilibrium efficiency between the samples and adsorbents. The effects of the extraction time were examined by measuring the extraction recoveries of the three flavonoids with methanol solutions of the tailor-made NADES Bet–Mal over the range 5–115 min. In the initial process, extraction of the three analytes was observed in the first 35 min (Figure 5a). No further increase in extraction efficiency was noted when the extraction period was prolonged. Therefore, the extraction capacity of Bet–Mal could be analyzed within 35 min.

Temperature also affects the extraction recoveries of the three flavonoids. Therefore, the extraction ability of Bet–Mal from 0 to 60 °C was studied with methanol as the solvent and the extraction time was set to 35 min. The maximum extraction recovery for quercetin, isorhamnetin, and kaempferol was 88.76%, 79.12%, and 80.41% at 20 °C (Figure 5b). At higher temperatures, the extraction recoveries for the three flavonoids were relatively constant. Therefore, a methanol solution of Bet–Mal at 20 °C was applied to extract quercetin, isorhamnetin, and kaempferol from the onion and broccoli samples.

The amount of NADESs is also another factor that affects the extraction recoveries of the three flavonoids. In this assay, five different amounts were tested: 50, 100, 200, 400, 800, and 1000 μL. Within the range tested, the recovery increased with increasing amounts added to the system, but no increase in recovery occurred when the amount of Bet–Mal added exceeded 400 μL (Figure 5c). Thus, 400 μL was chosen as the optimal amount. The inter-molecular hydrogen bonds become saturated when the amount of Bet–Mal was more than 400 μL.

### 2.5. Verification and Applications of the Method

The proposed method was evaluated from the aspects of the linear range (1–200 μg/mL), correlation coefficient, limit of detection (LOD), and limit of quantitation (LOQ) for the three flavonoids under the optimized extraction conditions. Excellent linearity was acquired with correlation coefficients greater than 0.9985, and the LODs for quercetin, isorhamnetin, and kaempferol were 0.16, 0.14, and 0.17 μg/g, respectively (Table 1).

The accuracy and reliability of the method was determined by conducting triplicate analysis of spiked samples and assessing the intraday and interday recoveries. The method recoveries for quercetin (Y = 4.2756X − 20.2, R^2^ = 0.99926), isorhamnetin (Y = 4.2444X − 36.884, R^2^ = 0.99853), and kaempferol (Y = 4.0205X − 18.683, R^2^ = 0.99931) were 88.91–98.99%, 88.45–99.01%, and 89.56–98.74% at concentrations of 10, 50, and 100 μg/mL (Table 2). The relative standard deviations for the intraday and interday detection were less than 5.01%. This suggests that the present method is reproducible and can be used for the analysis of analytes in real samples. The real quantities of quercetin, isorhamnetin, and kaempferol extracted from onion and broccoli with Bet–Mal added to methanol in the ultrasonic assisted solid liquid extraction method were 193.92, 136.25, and 98.47, and 197.54, 127.36, and 103.49 mg/kg, respectively.

### 2.6. Data Analyses

The means of three independent experiments (performed in triplicate) were analyzed using one-way analysis of variance (ANOVA) to ascertain the differences between the real samples and a Tukey’s multiple-range test was performed (*p* < 0.05).

### 2.7. Comparison with Other Reported Methods

The performance of this proposed method was evaluated by a comparison with different extraction methods developed with HPLC in terms of the LOD, LOQ, average recovery, and linear range (Table 3) [32,33,34,35].

## 3. Experimental

### 3.1. Chemicals and Reagents

Choline chloride (ChCl, Mw = 139.62 g/mol), betaine (Bet, Mw = 117.15 g/mol), meso-erythritol (Ery, Mw = 122.12 g/mol), d-(−)-ribose (Rib, Mw = 150.13 g/mol), xylitol (Xyl, Mw = 152.15 g/mol), l-(−)-fucose (Fuc, Mw = 164.16 g/mol), d-(+)-glucono-1, 5-lactone (Glul, Mw = 178.14 g/mol), d-(+)-mannose (Man, Mw = 180.16 g/mol), d-(+)-galactose (Gal, Mw = 180.16 g/mol), d-mannitol (Mal, Mw = 182.17 g/mol), d-glucuronic acid (Glua, Mw = 194.14 g/mol), methanol, nylon filter, and acetic acid (HAc, Mw = 60.05 g/mol) were purchased from Aldrich (Seoul, Korea). Chromatographic grade acetonitrile was supplied by Sigma Chemical Co. (St. Louis, MO, USA). The standard quercetin (Que, Mw = 302.24 g/mol), >99.0% (HPLC), isorhamnetin (Iso, Mw = 316.26 g/mol), and kaempferol (Kae, Mw = 286.23 g/mol) were obtained from Tokyo Chemical Industry Co., Ltd. (Tokyo, Japan). Ultrapure water used throughout the study was acquired from a Pure Power water purification system of Human Corporation (Seoul, Korea). Onion and broccoli were bought from a local market (Incheon, Korea). Unless stated otherwise, all reagents utilized in the experiment were of analytical reagent grade and used as received.

### 3.2. Instrumentation and Conditions

High performance liquid chromatography (HPLC) was conducted on a Waters 600s multisolvent delivery system with a Waters 1515 liquid chromatography unit (Waters, MA, USA) and a variable wavelength 2489 UV dual channel detector. Empower™ 3 software (Waters, MA, USA) was used to control the system as well as to acquire and analyze the data. The analysis was performed on an OptimaPak C18 column (5 μm, 250.0 × 4.6 mm, i.d., RStech Corporation, Daejeon, Korea). The injection loop volume was 20 μL. Sonic Power (Mirae Ultrasonic Tech. Co, Osaka, Japan) was used for the ultrasonic-assisted solid liquid extraction experiments (100 W, 20 kHz). HPLC detection was carried out using 0.2% acetic acid (in H_2_O): acetonitrile (65:35, *v*/*v*) at a flow rate of 0.8 mL/min. The injection volume was 10 μL, and the ultraviolet-visible detector wavelength was fixed to 285 nm.

Fourier transform infrared (FTIR, Vertex 80 V Bruker, Billerica, MA, USA) spectroscopy was performed using KBr disks at 4000–500 cm^−1^. The thermal stabilities of the chemicals used to prepare the NADESs were determined by thermogravimetric analysis (TGA, TG 209 F3 Tarsus, Netzsch, Selb, Germany) under a nitrogen atmosphere. The temperature was increased from 50 to 400 °C at a heating rate of 10 °C/min to determine the decomposition temperatures. Because they are very hygroscopic compounds, each substance was heated to 100 °C for 20 min before the measurement to avoid the effects of water on the decomposition curves. The viscosity of the NADESs was tested with an 1835 Uhler viscometer (Loikaw Instrument Co. Ltd., Shanghai, China) in a digital thermostatic bath WHB-6 (Daihan Scientific Co. Ltd., Seoul, Korea) at 25.0 °C.

### 3.3. Preparation of Choline Chloride-Based NADESs

Nine types of choline chloride-based deep eutectic solvents were prepared using the heating method according to previous work [36]. Briefly, choline chloride-meso-erythritol (ChCl-Ery), choline chloride-d-(−)-ribose (ChCl-Rib), choline chloride-xylitol (ChCl-Xyl), choline chloride-d-(+)-glucono-1, 5-lactone (ChCl-Glul), choline chloride-d-glucuronic acid (ChCl-Glua), choline chloride-l-(−)-fucose (ChCl-Fuc), choline chloride-d-mannitol (ChCl-Mal), choline chloride-d-(+)-mannose (ChCl-Man), and choline chloride-d-(+)-galactose (ChCl-Gal) were placed in a 100 mL round flask. The mixtures were stirred and heated in a water bath at 80.0 °C until a clear liquid formed. Appendix A illustrates the synthetic protocol, and Appendix A lists the variety and ratio of NADESs.

### 3.4. Preparation of Betaine-Based NADESs

The betaine-based NADESs were obtained by heating different carbohydrate-based HBDs and betaine to 80.0 °C with constant stirring until a homogeneous liquid formed. Appendix A lists the abbreviations of the NADESs produced.

### 3.5. Preparation of the Extraction Samples

Standard stock solutions were prepared by dissolving quercetin, isorhamnetin, and kaempferol separately in methanol at 1.0 mg/mL, which were then stored at −20 °C. Standard working solutions were prepared by diluting the stock solutions with the mobile phase to produce standard solutions at concentrations of 10, 25, 50, 100, and 200 μg/mL. The standard curves of quercetin, isorhamnetin, and kaempferol were linear after assaying five data points in duplicate (Appendix A).

Before grinding the samples to powder, onion and broccoli were cut into small and even pieces, and dried in an oven (40 °C). One gram of each powder sample was added to methanol, and extraction was performed using an ultrasonic machine at 20 °C for 35 min at a solid/liquid ratio of 1:30 g/mL, followed by filtering the suspension through a 0.45 μm nylon filter to obtain the extraction sample. The subsequent sample solution was then used further for ultrasonic-assisted solid liquid extraction and HPLC assays. The extraction yield was calculated according to the analyte levels determined by high performance liquid chromatography-ultraviolet detection (HPLC-UV) using the following equation: extracted quantity = (mass of analyte, mg)/(mass of weighed sample powder, g).

## 4. Conclusions

An efficient and green extraction method using NADESs was developed for the extraction of quercetin from onion and broccoli. The current study provided a practical example of NADESs as designer solvents to extract the bioactive compounds from the vegetables effectively and selectively. Among the 18 types of NADESs assessed, which were formed from a mixture of inexpensive, natural, and nontoxic components, Bet–Mal provided the highest extraction efficiency for the extraction of quercetin, kaempferol, and isorhamnetin. Subsequent optimization of the operational conditions improved the extraction recoveries up to 98.99% for quercetin, which was significantly higher than that from conventional stirring or heating extraction methods. Overall, NADESs are designer solvents that can be applied as sustainable and safe extraction media for biochemical applications.

## Figures and Tables

**Figure 1 molecules-24-02300-f001:**
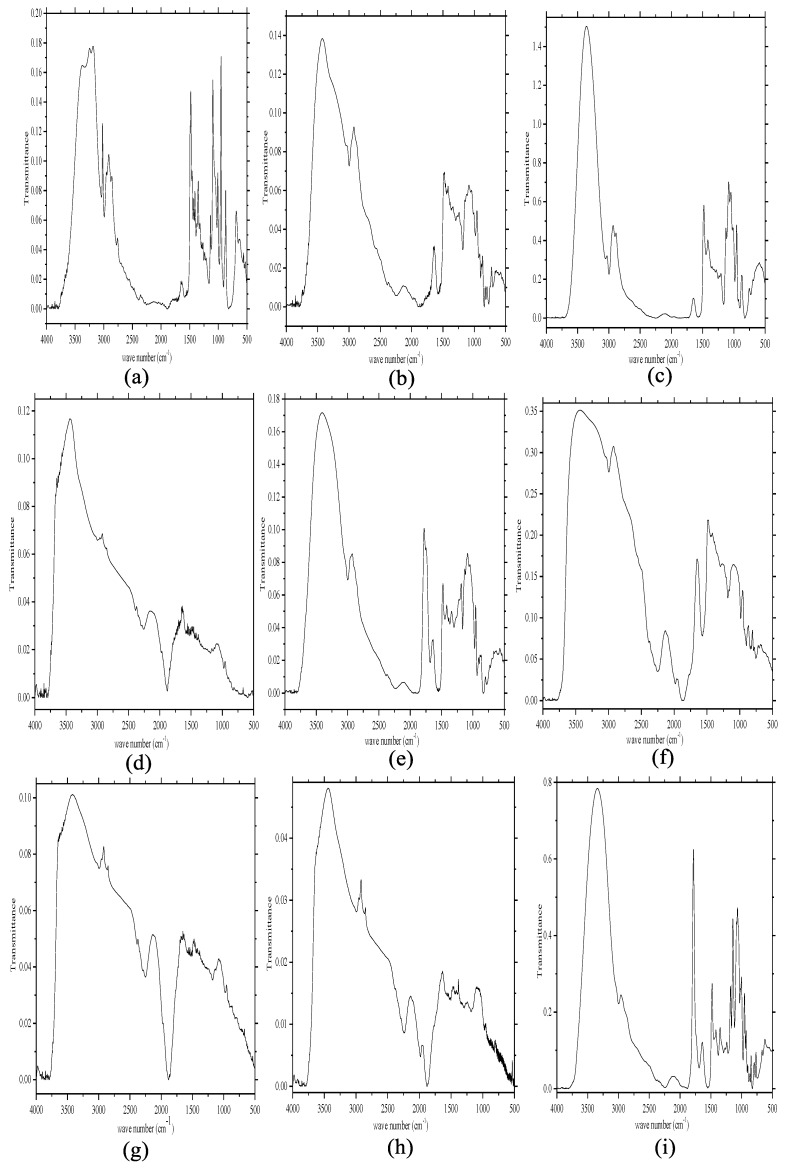
FT-IR spectra of ChCl-based NADESs. **a**: ChCl–Ery, **b**: ChCl–Rib, **c**: ChCl–Xyl, **d**: ChCl–Fuc, **e**: ChCl–Glul, **f**: ChCl–Man, **g**: ChCl–Gal, **h**: ChCl–Mal, and **i**: ChCl–Glua.

**Figure 2 molecules-24-02300-f002:**
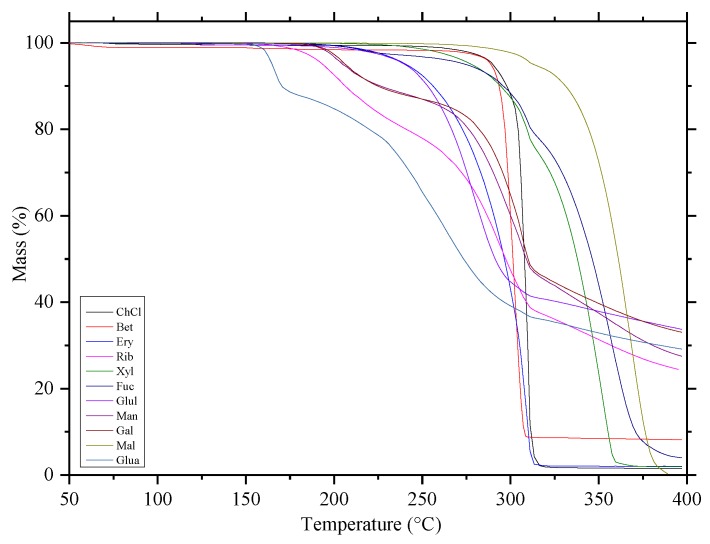
TGA curves of all the components.

**Figure 3 molecules-24-02300-f003:**
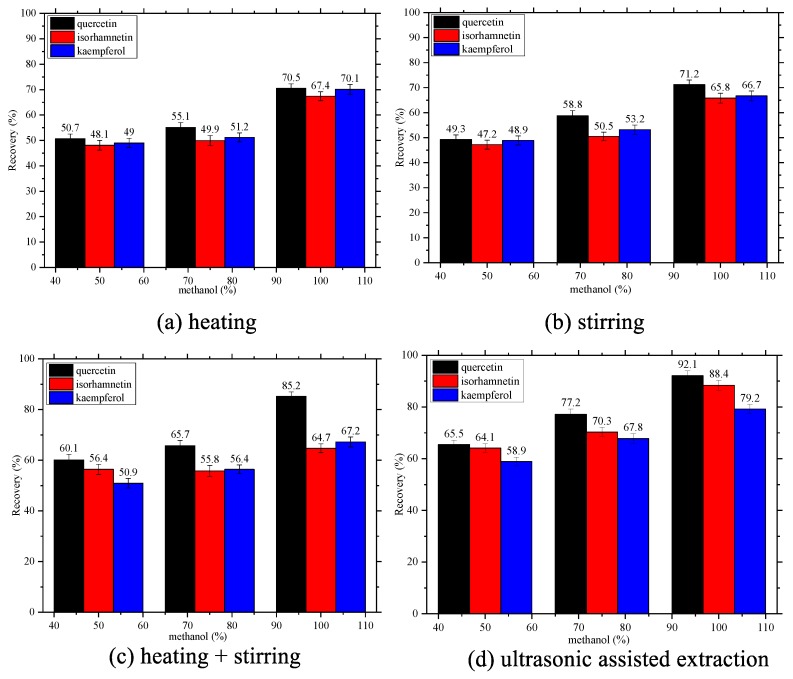
Selection of the extraction method and solvent.

**Figure 4 molecules-24-02300-f004:**
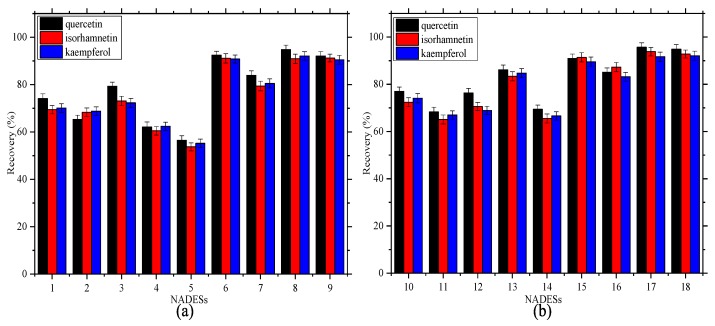
Selectivity of all the prepared NADESs. **a**: ChCl-based NADESs, and **b**: Bet-based NADESs.

**Figure 5 molecules-24-02300-f005:**
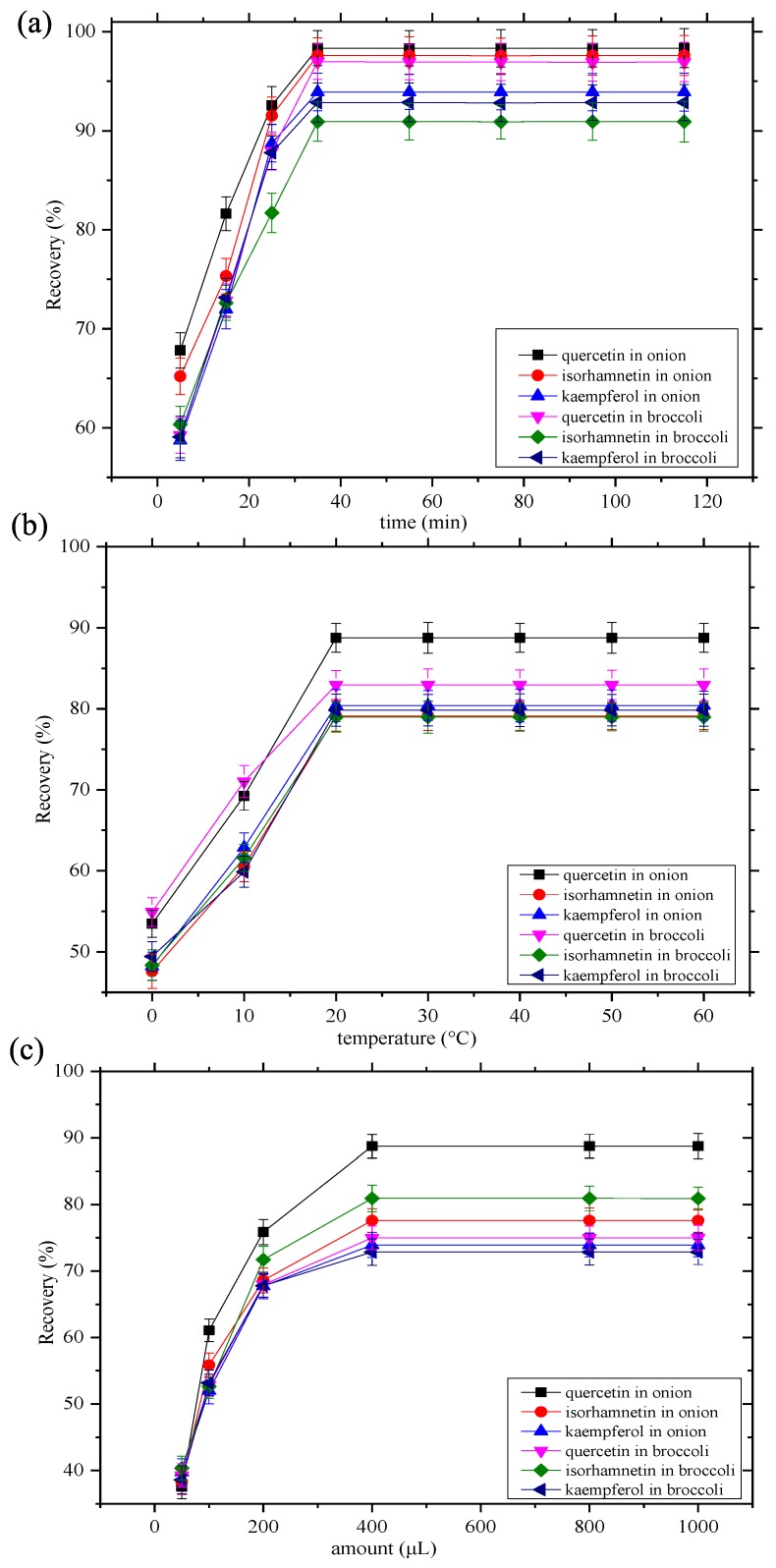
Optimization of the extraction conditions. **a**: effects of extraction time, **b**: effects of temperature, and **c**: effects of amount of NADESs added.

**Table 1 molecules-24-02300-t001:** Linearity, sensitivity and system precision (n = 9) of the proposed method.

Analyte	Range (μg/mL)	Equation	R^2^	LOD (μg/mL)	LOQ (μg/mL)
quercetin	1–200	Y = 4.2756X − 20.2	0.99926	0.16	0.51
isorhamnetin		Y = 4.2444X − 36.884	0.99853	0.14	0.43
kaempferol		Y = 4.0205X − 18.683	0.99931	0.17	0.52

**Table 2 molecules-24-02300-t002:** Recovery of the spiked standard solutions.

Analyte	Spike Concentration (μg/mL)	Inter-Day		Intra-Day	
		Recovery (%)	Mean ± Standard Deviation (%, n = 3)	Recovery (%)	Mean ± Standard Deviation (%, n = 3)
quercetin	10	85.23	85.23 ± 2.41	88.91	88.91 ± 5.01
	50	90.54	90.54 ± 4.03	90.02	90.02 ± 4.68
	100	98.76	98.76 ± 3.75	98.99	98.99 ± 3.78
isorhamnetin	10	84.17	84.17 ± 3.48	88.45	88.45 ± 4.32
	50	90.35	90.35 ± 2.15	92.13	92.13 ± 4.02
	100	97.92	97.92 ± 3.82	99.01	99.01 ± 4.33
kaempferol	10	84.52	84.52 ± 3.95	89.56	89.56 ± 3.95
	50	89.73	89.73 ± 2.51	91.05	91.05 ± 3.89
	100	98.02	98.02 ± 3.78	98.74	98.74 ± 4.06

**Table 3 molecules-24-02300-t003:** Comparison with other reported methods.

Method	Detection System	Extraction Solvent	Linear Range (μg/mL)	R^2^	LOD (μg/mL)	LOQ (μg/mL)	Average Recovery (%, n = 3)	References
ultrasonic- assisted solid liquid extraction	HPLC	methanol	1–200	0.99926	0.18	0.51	92.64	This study
solid liquid extraction		Methanol + H_2_O (0.1% *v/v* formic acid)	1–300	0.9993	0.33	1.06	99.2	[32]
ionic liquid-based cloud-point extraction		25% HCl	1–20	0.998	0.002	-	92.5	[33]
rat plasma extraction		rat plasma	0.1–25	0.9941	0.05	0.1	-	[34]
ultrahigh pressure extraction		60% methanol	0.005–0.05	0.9994	0.039	0.13	98.8	[35]

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
