# Peer review of "Application of Natural Deep Eutectic Solvents in the Extraction of Quercetin from Vegetables"

_molecules, 2019, doi:10.3390/molecules24122300_

Round 1

Reviewer 1 Report

In introduction section, 31 line, the authors should discuss more about the environmentally sustainable extraction methods reported in the literature and the importance of the use green solvents (Capello et al.Green Chem., 2007, 9, 927-934). 
Several extraction methods that use eco-sustainable solvents to obtain nutraceuticals  (Procopio et al. Journal of Agricultural and Food Chemistry, 2009, 57(23), 11161-11167; Nardi et al. Food Chemistry, 2014,162, 89-93; Nardi et al. Food and Function, 2017, 8(12),001) and target molecules obtained by biomass (Nardi et al. Green Chem., 2017, 19, 5403-5411) are reported.I believe that the manuscript can be accepted after a major revision in the introduction section giving a greater attention to the use of green solvents in the biomass field.

Author Response

We discussed more about the environmentally sustainable extraction methods reported in the literature and the importance of the use green solvents (Capello et al. Green Chem., 2007, 9, 927-934) in the introduction part as suggested.

Several extraction methods that use eco-sustainable solvents to obtain nutraceuticals  (Procopio et al. Journal of Agricultural and Food Chemistry, 2009, 57(23), 11161-11167; Nardi et al. Food Chemistry, 2014,162, 89-93; Nardi et al. Food and Function, 2017, 8 (12), 001) and target molecules obtained by biomass (Nardi et al. Green Chem., 2017, 19, 5403-5411) are reported. And we have added these references to the introduction part as suggested as references 12, 22, 23, 29, and 30.

Reviewer 2 Report

The Manuscript ID: molecules-535394 reports results of the study “Application of Natural Deep Eutectic Solvents in the Extraction of Quercetin from Vegetables”. These results may be interesting for people dealing with disease prevention and health promotion activities that have attracted significant research attention.  I recommend this manuscript for publication in this journal after minor revision.

The authors used carbohydrate based DESs. These types of DESs should be highly viscous which very difficult to use in the extraction. Therefore, the physical properties should be provided of these DESs in the revised version.

Please show the effect of water on the extraction.

The citation about DESs is not enough, some very related references to this work should be cited, such as (1) Mohammad Chand Ali, Jia Chen, Haijuan Zhang, Zhan Li, and Hongdeng Qiu, Effective extraction of flavonoids from Lycium barbarum L. fruits by deep eutectic solvents-based ultrasound-assisted extraction, Talanta, 2019, 203, 16-22; (2) Mohammad Chand Ali, Ruirui Liu, Jia Chen, Tianpei Cai, Haijuan Zhang, Zhan Li, Honglin Zhai, Hongdeng Qiu, New deep eutectic solvents composed of crown ether, hydroxide and polyethylene glycol for extraction of non-basic N-compounds, Chinese Chemical Letters, 2019, 30, 871-874.

Author Response

Thank you very much for your approval.

The viscosity of these natural deep eutectic solvents was performed using a 1835 Uhler viscometer (Loikaw Instrument Co. Ltd, Shanghai, China) in a digital thermostatic bath WHB-6 (Daihan Scientific Co. Ltd., Korea) at 25.0 °C. The results were as shown in Table S1.

We tested 50 %, 75 %, and 100% methanol for the extraction assays, and the results indicated 100% methanol to be the most effective. Therefore, 100% methanol was used as the extraction solvent to compare the extraction recoveries of various natural deep eutectic solvents in the following assays. The results were illustrated in Figure 3. Water showed a negative effect on the extraction of quercetin, as the recovery increased with the decrease of water percent.

These two related references have been added to this work as references 26 and 27.

Reviewer 3 Report

This MS describe a new method for the extraction of quercetin from vegetables based on the use of natural deep eutectic solvents (NADES). The subject of this MS is very interesting and innovative. However, I think that the abstract needs to be completely revised since the most important concept (i.e., the use of NADES to extract quercetin) is not present.

Another little observation is at line 207, p. 10: is the 0.2% acetic acid in water? Please specify this.

Author Response

The abstract was completely revised by addition of the use of NADES to extract quercetin as suggested.

We have specified this as suggested in the text, 0.2 % acetic acid was prepared by adding 2 mL acetic acid to 1000 mL H2O. Then it was further mixed with acetonitrile at a ratio of 65:35 (v/v) and used as the mobile phase.
